# Peptides for Coating TiO_2_ Implants: An In Silico Approach

**DOI:** 10.3390/ijms232214048

**Published:** 2022-11-14

**Authors:** Almerinda Agrelli, Niedja Fittipaldi Vasconcelos, Rayane Cristine Santos da Silva, Carina Lucena Mendes-Marques, Isabel Renata de Souza Arruda, Priscilla Stela Santana de Oliveira, Luzia Rejane Lisbôa Santos, Audrey Nunes de Andrade, Ronald Rodrigues de Moura, Lucas Coelho Bernardo-Menezes, Natália Pereira da Silva, Giovanna Machado

**Affiliations:** 1Laboratory of Nanostructured Materials (LMNANO), Strategic Technologies Center of Northeast (CETENE), Recife 50740-545, Brazil; 2Department of Advanced Diagnostics, IRCCS Materno Infantile Burlo Garofolo, 34137 Trieste, Italy; 3Laboratory of Virology and Experimental Therapeutics (LaViTE), Aggeu Magalhães Institute, Oswaldo Cruz Foundation (FIOCRUZ), Recife 50670-420, Brazil; 4Northeast Biotechnology Network (RENORBIO), Postgraduation Program in Biotechnology, Federal University of Pernambuco, Recife 50670-901, Brazil

**Keywords:** TiO_2_ implants, osseointegration, titanium, molecular docking, molecular dynamics

## Abstract

Titanium is usually used in the manufacturing of metal implants due to its biocompatibility and high resistance to corrosion. A structural and functional connection between the living bone and the surface of the implant, a process called osseointegration, is mandatory for avoiding prolonged healing, infections, and tissue loss. Therefore, osseointegration is crucial for the success of the implantation procedure. Osseointegration is a process mediated by bone-matrix progenitor cells’ proteins, named integrins. In this study, we used an in silico approach to assemble and test peptides that can be strategically used in sensitizing TiO_2_ implants in order to improve osseointegration. To do so, we downloaded PDB structures of integrins α5β1, αvβ3, and αIIbβ3; their biological ligands; and low-cost proteins from the Protein Data Bank, and then we performed a primary (integrin-protein) docking analysis. Furthermore, we modeled complex peptides with the potential to bind to the TiO_2_ surface on the implant, as well as integrins in the bone-matrix progenitor cells. Then we performed a secondary (integrin–peptide) docking analysis. The ten most promising integrin–peptide docking results were further verified by molecular dynamics (MD) simulations. We recognized 82 peptides with great potential to bind the integrins, and therefore to be used in coating TiO_2_ implants. Among them, peptides 1 (GHTHYHAVRTQTTGR), 3 (RKLPDATGR), and 8 (GHTHYHAVRTQTLKA) showed the highest binding stability during the MD simulations. This bioinformatics approach saves time and more effectively directs in vitro studies.

## 1. Introduction

Since the discovery of osseointegration in 1950 by Per-Ingvar Brånemark, several studies have been carried out with the objective of improving the fixation between the bone tissue and the TiO_2_ implant [1]. Osseointegration is seen as biofunctional implant fixation and bone apposition [2].

Among the different types of materials for implants currently known, titanium and its alloys provide the best biocompatibility due to their unique physicochemical and biological properties, such as good mechanical properties, high corrosion resistance, stability in body fluids, and low cytotoxicity [3,4]. Its high biocompatibility is due to the titanium dioxide (TiO_2_) layer, which is formed on the titanium surface when it comes into contact with oxygen [5,6].

Currently, six different types of titanium are available as implant biomaterials. Among them, four are grades of commercially pure titanium (CPTi) (Grade I, Grade II, Grade III, and Grade IV) and two are titanium alloys (Ti-6Al-4V and Ti-6Al-4V-Extra Low Interstitial alloys) [7].

An ideal material for the fabrication of implants should be biocompatible and have adequate strength, toughness corrosion, and fracture resistance. These properties are usually related to the oxygen residuals in the metal. Grade IV CpTi presents the highest oxygen content (0.4%) and, consequently, excellent mechanical strength, and that is why it is the most widely used type of titanium for implants [8].

Fixation between the titanium device and the patient’s bone tissue is very important for the long-term success of the implant [9]. Otherwise, small gaps can form locally at the tissue–implant interface due to metabolic bone diseases or failures in the healing process (chronic or acute inflammatory reactions) at the implantation site [10].

After the bone-drilling procedure, the bleeding and bone debris generated by the trauma induce the release of several cytokines and bone-matrix proteins at the site of the implant, where they come into contact with the implant surface [11]. The hydrophilicity of the titanium surface, as well as the charge and topography, plays an important role in the initial healing, immediately favoring the adhesion of plasma proteins (albumin, for example), which are simultaneously replaced by other proteins with higher affinity, such as fibronectin and vitronectin, forming a platelet plug and, consequently, leading to tissue hemostasis [12,13,14,15]. After that, thrombocytes cells promote the degranulation of the platelet plug, releasing growth factors and cytokines, which induce the inflammatory phase [13]. At this critical stage, neutrophils and mononuclear cells are present around the dental implant [16].

Subsequently, the macrophages adhere to the implant surface and fuse, forming foreign-body giant cells. These cells stimulate fibroblasts to initiate fibrogenesis [16,17]. At this stage, the implant acquires primary stability due to the collagen filling by fibroblasts in the bone–implant interface. Simultaneously, other proteins and cytokines are secreted by macrophages, thus leading to the angiogenesis and differentiation of mesenchymal stem cells into osteoblasts [13,16]. Bone morphogenetic proteins are released from the old bone matrix, conducting the formation of new bone around the implant surface [13,18]. Therefore, the osseointegration process encompasses the first three phases of the healing process: hemostasis, immunoinflammatory, and osteogenesis/angiogenesis [19]. On a molecular level, these phases occur progressively and are mediated by glycoproteins present in the extracellular matrix, such as fibronectin, vitronectin, and fibrinogen. These proteins bind to bone-matrix progenitor cells’ receptors, called integrins, such as α5β1, αvβ3, and αIIbβ3 [20,21,22]. Integrins αvβ3 and α5β1 are involved in the mechanisms of cell adhesion and differentiation and induce osteogenesis [23], while αIIbβ3 is required for cell-to-cell interactions and platelet aggregation [24].

To prevent complications in the osseointegration process, several methodological approaches have been sought in order to control the inflammatory phase and stimulate the process of osteogenic cell adhesion, osteoblastic differentiation, and bone-formation activity [25]. In this context, treating the implant surface with adhesion proteins or functional peptides could accelerate osseointegration at the bone–implant interface and favor the osteogenesis process [10]. Considering this, peptides have lower antigenicity, greater selectivity, and greater chemical immobilization capacity on the implant surface [10,25]. Thus, peptides that present a biofunctional sequence have the potential to alter the local environment for biologically favorable circumstances, and this may be important for patients with bone or systemic metabolic diseases [26].

Functionalizing implants with bioactive molecules with potential for recognition by integrins may promote faster and stronger osseointegration. Thus, a promising integrin-binder should present similar proprieties to extracellular matrix proteins (the integrins’ biological ligands). The cell-to-cell adhesion oligopeptide RGD (arginine—glycine—aspartic acid) is a specific site in the integrins’ ligands that is recognized by most integrins [27], but it has already been widely studied. Furthermore, most of the assays conducted with peptide sensitization of TiO_2_ consider only the affinity of the peptide with integrins or with TiO_2_. We failed to find studies with peptides in the literature that presented affinity properties for both.

In this work, we aimed to construct peptides for sensitizing TiO_2_ implants and help improve osseointegration with two functionalities: one domain binding to TiO_2_ and another displaying high binding affinity for integrins. To achieve the latter, we identified peptides from integrins’ biological ligands. After that, we tested these peptides through molecular docking and dynamics simulations. This cost-efficient and time-saving type of approach can provide insights and guide researchers to develop more precise hypotheses.

## 2. Results

The main objective of this work was to test peptides that could serve to sensitize TiO_2_ implants to increase osseointegration. It is important to note that, in addition to those discussed in this work, other integrins participate in the osseointegration and osteogenesis processes. Therefore, a limitation of this work was the availability of PDB structures of the integrins.

The 52 modeled motifs/peptides are shown in Appendix A. Moreover, we identified 82 peptides with higher affinity binding than the RGD motif (Appendix A).

The ten peptides that showed stronger binding affinity with each integrin are presented in Table 1. As integrin α5β1 is expressed by bone-marrow stromal cells, osteoblasts, and osteoprogenitors cells in different stages of osteogenesis, it plays a key role in osteogenic differentiation and osseointegration [22]. For that reason, we chose to perform a further molecular dynamic analysis of the three studied integrins with the ten peptides that presented stronger affinity binding with integrin α5β1. We named the peptides (pep1–10) in descending order of affinity binding (Table 2).

All plots for RMSD, Rg, SASA, and H-bounds analyses can be found in the Appendix A. Based on the trajectory analyses, Figure 1 displays the three best complexes. The criteria for selection were based on the best combination of binding stability, solvent surface accessibility, less conformational changes, and number of formed hydrogen bonds between the receptor and ligand.

## 3. Discussion

The integrin family is composed of 24 heterodimeric cell surface transmembrane proteins that mediate cell interactions with the extracellular matrix [28]. These proteins are formed by two subunits: α and β. To date, 18 α and 8 β subunits have been identified. Of the 24 different integrins, only 8 recognize the RGD sequence in their biological ligands. Among them are integrins α5β1, αvβ3, and αIIbβ3 [29,30]. The RGD motif is found in many extracellular matrix molecules, such as fibronectin, vitronectin, bone sialoprotein, and osteopontin [31].

The α5β1 integrin is expressed by osteoblasts and also by bone marrow stromal cells [32]. It promotes osteogenic differentiation, cell survival and matrix mineralization [33,34]. In fact, it has been reported that a decrease in α5β1–fibronectin interaction leads to apoptosis [35]. Although few α5β1antagonists are currently known [36], this integrin is reported to mediate cell proliferation, angiogenesis, and metastasis and is closely linked to the progression of several types of cancer [37]. In fact, the inhibition of integrins has been a major challenge in the development of anticancer therapy [38]. According to the docking analysis, the peptides GHTHYHAVRTQTTGR, RKLPDALKA, and RKLPDATGR demonstrated the stronger binding affinity with α5β1 integrin.

In contrast, peptides RKLPDALKA, RKLPDATGR, and QPYLFATDSLIKLKA showed the highest binding affinity with the αvβ3 integrin. Integrin αvβ3 is expressed by osteoclasts and plays a role in cellular adhesion and resorption [39,40]. Moreover, integrin αvβ3 is widely studied because of its important role in angiogenesis and tumorigenesis. In this context, a variety of cancer treatment strategies may be used by targeting this integrin, such as the development of αvβ3 antagonists or RGD conjugates to deliver an anticancer drug system [41,42].

Integrin αIIbβ3 is expressed on platelets, megakaryocytes, basophils, mast cells, and some tumor cells [24]. When activated, it mediates platelet aggregation by serving as a receptor for ligands that can bridge to other αIIbβ3s on adjacent platelets. This mechanism is responsible for bridging platelets together [24]. Integrin αIIbβ3 antagonists are used in the clinic to inhibit platelet aggregation and prevent thrombosis [43]. The peptides GHTHYHAVRTQTTGR, RKLPDATGR, and QPYLFATDSLIKLKA showed the stronger binding affinity with integrin αIIbβ3 in the docking analysis.

The molecular dynamic analyses were performed with the ten peptides that showed the best results in the docking analysis with integrin α5β1 for reasons previously described. Considering the four parameters (RMSD, SASA, Rg, and H-bonds) used in the molecular dynamic analysis, the peptides that demonstrated the greater complex stability with the three integrins were pep1 (Figure 1A), pep3 (Figure 1B), and pep8 (Figure 1C) (the PDB structures of these complexes are available in Appendix A). Furthermore, integrin αIIbβ3 showed the highest binding stability with all the ten analyzed peptides, suggesting that this integrin may present low specificity and might represent an interesting target for sensitized implants.

It is important to note that pep1 (GHTHYHAVRTQTTGR) showed the best binding affinity when docked with the integrins α5β1 and αIIbβ3, and it showed the fourth best when docked with integrin αvβ3. The structure of the integrins (α5β1, αvβ3, and αIIbβ3) alone and in complex with peptide 1 after docking and molecular dynamics simulations are shown in Figure 2. To better observe the interactions between residues, LIGPLOTS were generated by using LigPlot+ v.4.5.3 [44].

In addition, the molecular dynamic analysis demonstrated that pep1 formed the strongest complex stability with the three integrins, although the results between pep1, pep3, and pep8 were very similar. Interestingly, the TiO_2_-binding sequence GHTHYHAVRTQT is present in pep1 and pep8, while the integrin-binding sequence TGR is present in pep1 and pep3. These findings suggest that pep1, pep3, and pep8 are the best candidates for coating TiO_2_ implants.

As mentioned before, the surface characteristics of the implant affect cell adhesion, proliferation, and osteoblast differentiation, ultimately dictating the osseointegration process and the success rate of implants. Because of RGD’s potent binding, it has been widely studied as an adhesive ligand. However, several studies determining implant fixation have shown that the use of RGD peptides onto implant surfaces fails to enhance functional osseointegration, including studies conducted on titanium implants which demonstrated that RGD-coated implants cause no improvements in osseointegration [45,46,47,48].

In this context, modifying the topography of the TiO_2_ implant at a nanoscale has been shown to increase cell adhesion and differentiation in MC3T3-E1 cells and upregulate the expression of integrins α10, α11, α7, β3, β5, and αV, which are involved in osteoblast differentiation [49]. In accordance with this, different types of rough topography seem to have a positive influence on the cell morphology in MG-63 osteoblast cell culture [50]. Furthermore, cells exposed to a rough titanium surface seem to be more resistant to breakdown when processed for scanning electric microscopy analysis, suggesting that rough titanium surfaces induce a stronger mode of attachment [51]. Additionally, an in vitro study showed that SaOS-2 cells demonstrated early attachment and spreading in titanium treated with peptidomimetics (targeting the integrins α5β1 and αvβ3) compared to unmodified titanium. Moreover, an increased cell proliferation and mineralization were observed on the surfaces coated with the peptidomimetics [52]. These results suggest that, to enhance osseointegration, a titanium implant should present a rough topography that mimics the bone-tissue surface and an adhesive molecule that targets integrins in bone-matrix progenitor cells.

Concerning biomimetics, peptides have been extensively studied and seem to be promising molecules. A recent study found, through molecular docking and dynamics simulations, that a peptide (KRSSR) attaches to the TiO_2_ surface in a stable position [53]. Moreover, in vitro studies have been widely performed. Biomimetizing TiO_2_ with a single phosphorylated amino acid (threonine) has been demonstrated to enhance osseointegration in vitro and in vivo [54]. An adhesive peptide of arginine–glycine–aspartic acid–cysteine (RGDC) has also been immobilized onto anodized TiO_2_ nanotubes and tested in rat bone-marrow stromal cells. The results demonstrated increased adhesion and osteogenic gene expression [55].

More recently, using molecular docking and dynamics simulations, a study found two flavonoids to present high affinity binding to critical regions of the SARS-CoV-2 spike protein [56]. In vitro, these flavonoids were immobilized onto TiO2 nanoparticles and had their antiviral effect tested in two different coronaviruses (HCoV 229E and SARS-CoV-2). A clear antiviral effect was observed against the two viral strains. These convergent results reinforce the importance of in silico studies in directing bench assays. In addition, a similar approach to the one we used in this work was used to test antimicrobial peptides in vitro [57]. The chimeric peptide tested presented one domain with the capacity to bind titanium and another domain displaying an antimicrobial property. The results demonstrated that the complex peptides efficiently coated the titanium alloy surface, as well as provided an antimicrobial effect. Considering the promising previous trials with peptides, our results can guide future and ongoing in vitro trials.

## 4. Materials and Methods

The methodology that was used is summarized in Figure 3.

### 4.1. Proteins and Peptides Preparation

Firstly, we searched the literature to recognize integrins involved in the osseointegration process and their respective biological ligands. Furthermore, aiming to meet a commercial demand, we searched for low-cost proteins. The 3D structures of the integrins, (α5β1, αvβ3, and αIIbβ3), their biological ligands (fibronectin, vitronectin, and fibrinogen) and the low-cost proteins (Bovine Serum Albumin (BSA), Ovine Serum Albumin (OSA), and Myosin) were retrieved from the Protein Databank (PDB) [58]; the PDB IDs were 3VI3, 4MMX, 6V4P, 1FNA, 2JQ8, 3GHG, 4F5S, 5ORF, and 2VAS, respectively. Water and other molecules and ions were removed from the structures by using Chimera software version 1.13.1 package from the Resource for Biocomputing, Visualization, and Informatics at the University of California, San Francisco (supported by NIH P41 RR-01081) [59].

After that, using GalaxyPepDock web server (Seoul National University, Korea) [60], we modeled complex peptides with the potential to bind to the TiO_2_ surface on the implant, as well as integrins in the bone-matrix progenitor cells. To achieve this, we first identified the motifs (of three amino acids each) involved in the binding of integrins with their ligands (biological ligands and low-cost proteins), using the 3D files generated from the integrin–protein docking results. Then we searched the literature to determine peptides known to strongly bind TiO_2_. We found QPYLFATDSLIK, GHTHYMAVRTQT, and RKLPDA [61,62]. Afterwards, we assembled and modeled the motifs alone, and putted together with the TiO_2_-binding peptides (Appendix A). Model refinement and validation were performed by 3DRefine web server (University of Missouri, Columbia) [63] and PROCHECK (version 3.5.4, University College, London) [64] software, respectively. Next, we performed further docking simulations of these motifs and complex peptides with each one of the studied integrins.

### 4.2. Docking and Molecular Dynamics Simulations

Firstly, using ClusPro 2.0 web server (Boston University, Boston) [65], we performed docking simulations of the integrins with their biological ligands (intending to set a parameter of control) and with the low-cost proteins (intending to identify their binding motifs). The ClusPro 2.0 server measures the van der Waals force and electrostatic interactions (VdW + Elec) of protein–protein complexes. After that, and once we had assembled the complex peptides, we docked the integrins with these peptides. Altogether, we performed 18 protein–integrin docking simulations (Appendix A) and 156 peptide–integrin docking simulations (52 motifs/peptides with each of the three analyzed integrin) (Appendix A). The ten peptides with the lowest cluster score were chosen and submitted to molecular dynamics simulations.

The molecular dynamics simulations were carried out by using the GROMACS package (GROningen MAChine for Chemical Simulations) (version 2018.3, KTH Royal Institute of Technology, Stockholm) [66]. Force-field GROMOS 96 53a6 was used [67]. The system was soaked in water and ionically balanced, and system energy was minimized. During the phase of system equilibration, the temperature and pressure were set at 310K and 1 bar, respectively. Root-mean-square deviation (RMSD) patterns were observed as a function of the simulation time (100 ns) of each system, as well as the solvent accessible surface area (SASA). The distribution of hydrogen bonds (H-bonds) and the radius of gyration (Rg) of the polypeptide chain amino acids were also analyzed. The RMSD analysis was used to predict the stability of the protein. A lower RMSD value implies high overall stability of the protein structure. SASA is a parameter that measures the accessible surface area to solvent molecules. The rise in SASA value denotes relative expansion and, therefore, low compression/stability. Rg analyzes the dimensions of the protein. It stands for the mass weighted root-mean-square distance of a collection of atoms from their common center of mass. The smaller the distance, the more compacted/stable the complex. Finally, the H-bonds play an important role in the protein stability. The more H-bonds formed, the greater the force of interaction between the molecules of the complex [68].

## 5. Conclusions

To develop more biocompatible TiO_2_ implants, surface treatments are often used. Targeting the bone-matrix progenitor cells is an interesting strategy that can reduce healing time and prevent infections. Here, we used a bioinformatic approach to identify peptides with potential for coating TiO_2_ implants. We identified, through a docking analysis, 82 peptides with higher-affinity binding for the integrins α5β1, αvβ3, and αIIbβ3 than their biological ligand (RGD). The molecular dynamic simulations demonstrated that peptides 1 (GHTHYHAVRTQTTGR), 3 (RKLPDATGR), and 8 (GHTHYHAVRTQTLKA) showed the highest binding stability in complex with the tested integrins, and therefore, they are the best candidates for coating TiO_2_ implants. In addition to the interest in coating TiO_2_ implants, the results described here also serve as guidance to help researchers in other medical areas, such as in anticancer and thrombolytic drugs. Further in vitro studies are needed.

## Figures and Tables

**Figure 1 ijms-23-14048-f001:**
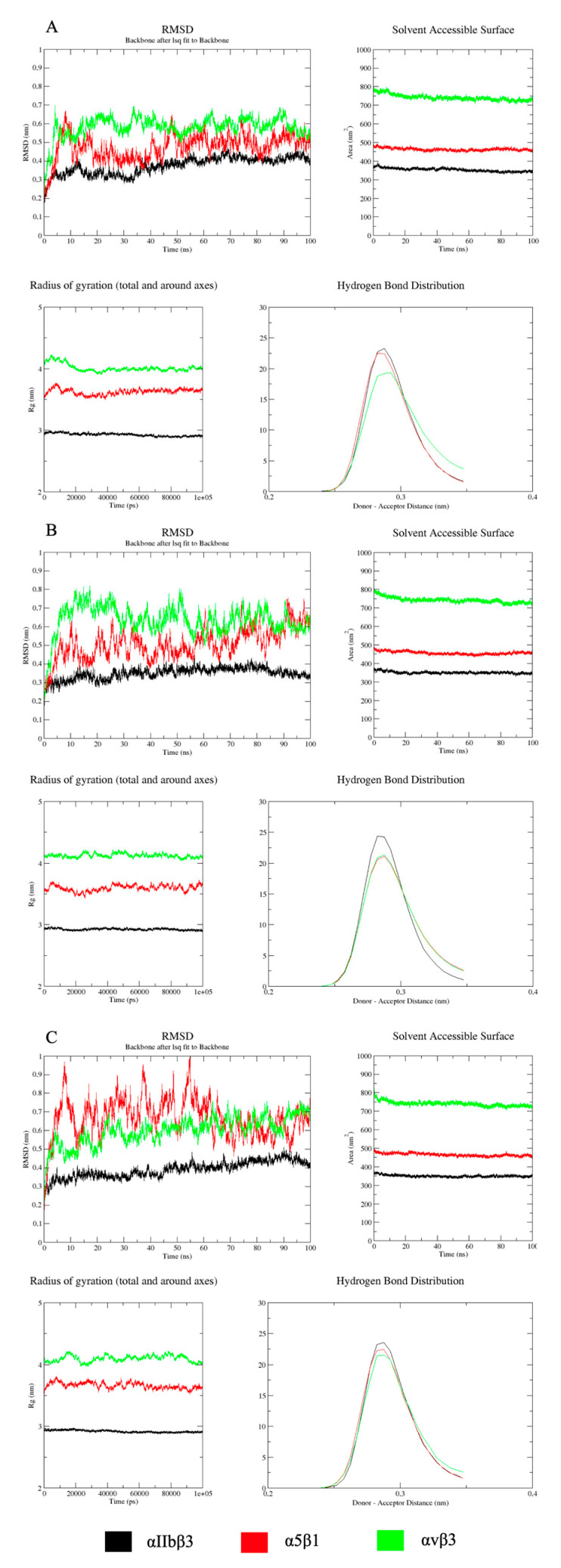
Pep1 (**A**) and pep3 (**B**) demonstrated the lowest RMSD, SASA, and Rg values and, therefore, the highest complex stability with integrins αIIbβ3, α5β1, and αvβ3, respectively. Considering pep8 (**C**), integrins α5β1 and αvβ3 demonstrated very similar RMSD values; however, SASA and Rg values showed that this peptide also demonstrated highest stability for integrins αIIbβ3, α5β1, and αvβ3, respectively.

**Figure 2 ijms-23-14048-f002:**
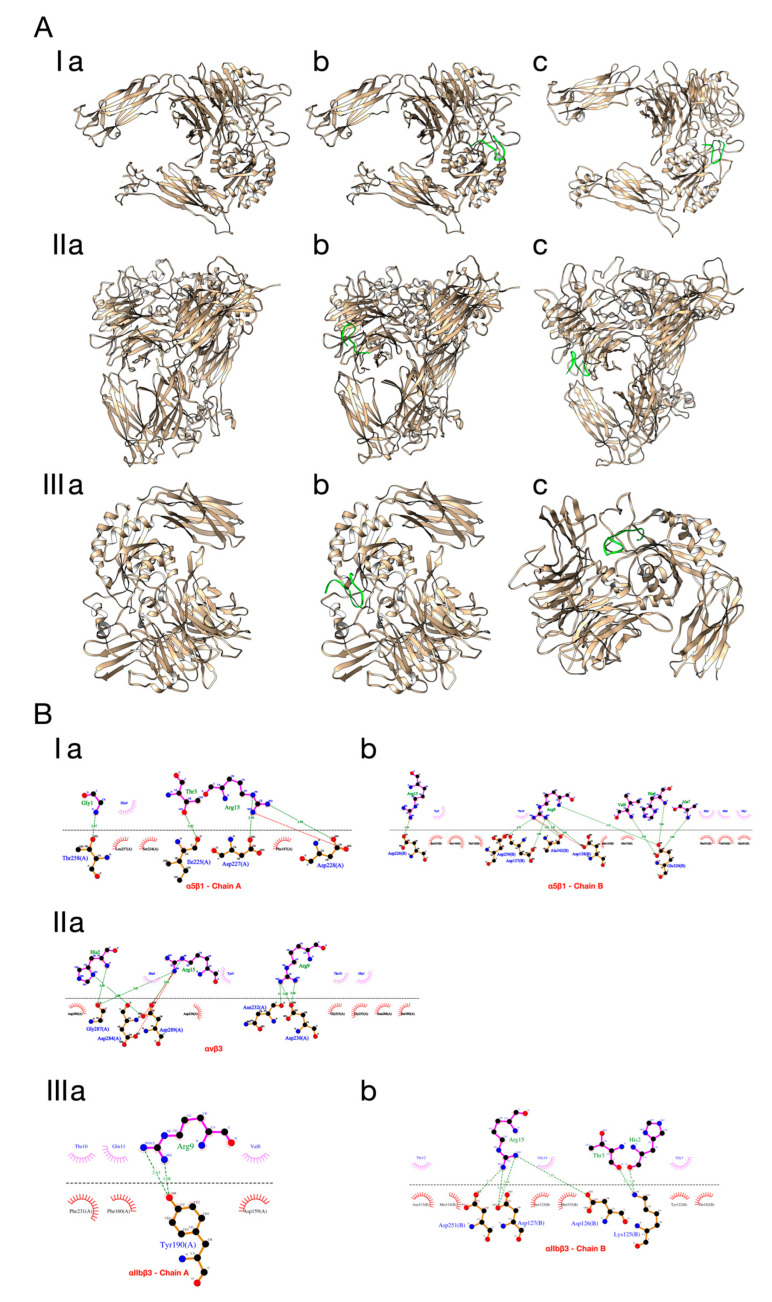
**A**(**I**) α5β1 alone (**a**), and in complex with peptide 1 after docking (**b**) and dynamics simulations (**c**); **A**(**II**) αvβ3 alone (**a**), and in complex with peptide 1 after docking (**b**) and dynamics simulations (**c**); **A**(**III**) αIIbβ3 alone (**a**), and in complex with peptide 1 after docking (**b**) and dynamics simulations (**c**); (**B**) LIGPLOTS of the binding interactions between peptide 1 and the integrins α5β1 (**I**), αvβ3 (**II**), and αIIbβ3 (**III**). In the LIGPLOTS, peptide 1 is above the dotted line, while integrins are below. The arcs with red spines represent hydrophobic interactions, while the green dotted lines indicate hydrogen bonds.

**Figure 3 ijms-23-14048-f003:**
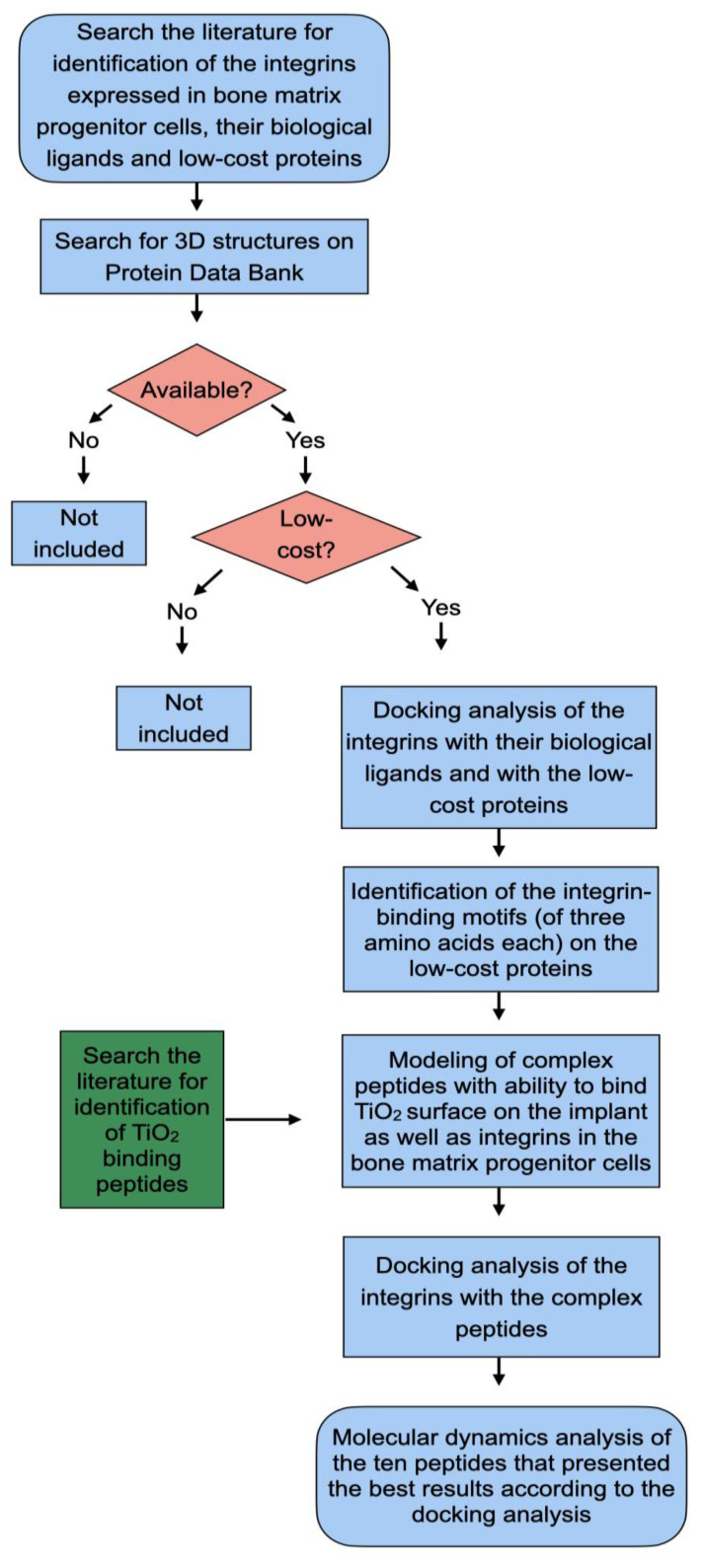
Flowchart of the study steps.

**Table 1 ijms-23-14048-t001:** Ten peptides that demonstrated the higher affinity binding for each integrin. The docking analysis measures the affinity binding of the integrin–peptide complexes by the analysis of van der Waals and electrostatic interactions (VdW + Elec). The lower the cluster score, the greater the affinity. The biological ligand of the integrins (RGD motif) is underlined and is meant to serve as a comparison parameter. The peptide that presented the higher affinity binding is highlighted in bold.

Integrin α5β1 (VdW + Elec)	Peptide Sequence	Integrin αvβ3 (VdW + Elec)	Peptide Sequence	Integrin αIIbβ3 (VdW + Elec)	Peptide Sequence
−155.5	RGD	−163.5	RGD	−150.4	RGD
−238.2	**GHTHYHAVRTQTTGR**	−234.4	**RKLPDALKA**	−242.6	**GHTHYHAVRTQTTGR**
−227	RKLPDALKA	−225.9	RKLPDATGR	−235.6	RKLPDATGR
−218.1	RKLPDATGR	−224	QPYLFATDSLIKLKA	−223.3	QPYLFATDSLIKLKA
−217.4	RKLPDARGD	−219	GHTHYHAVRTQTTGR	−221.9	RKLPDALKA
−215.4	QPYLFATDSLIKLKA	−216.3	RKLPDARGD	−213.7	GHTHYHAVRTQTLKA
−204.9	GHTHYHAVRTQTQAG	−213.4	GHTHYHAVRTQTDLN	−213.5	GHTHYHAVRTQTRGD
−204.2	GHTHYHAVRTQTRGD	−210.8	GHTHYHAVRTQTGVL	−202.2	GHTHYHAVRTQTDLN
−198.1	GHTHYHAVRTQTLKA	−208.8	GHTHYHAVRTQTPDG	−200	GHTHYHAVRTQTQAG
−196.4	GHTHYHAVRTQTDLN	−207.6	QPYLFATDSLIKKDD	−183.1	RKLPDAVLP
−195.9	RKLPDAVLP	−205.3	RKLPDAVLP	−182.6	GHTHYHAVRTQTGVL

**Table 2 ijms-23-14048-t002:** Docking scores (VdW + Elec) of the ten peptides that demonstrated the higher binding affinity for integrin α5β1 and their respective cluster scores for integrins αvβ3 and αIIbβ3. The peptides with better affinity for the integrin α5β1 were chosen for a further molecular dynamics analysis because of this integrin key role in the osteogenic process. The biological ligand of the integrins (RGD motif) is underlined and is meant to serve as a comparison parameter. Peptide 1 (pep1) showed the best binding affinity when docked with the integrins α5β1 and αIIbβ3, and it was the fourth best when docked with integrin αvβ3 (highlighted in bold).

	Peptide	Integrin		
		α5β1	αvβ3	αIIbβ3
	RGD	−155.5	−163.5	−150.4
Pep1	**GHTHYHAVRTQTTGR**	−238.2	−219	−242.6
Pep2	RKLPDALKA	−227	−234.4	−221.9
Pep3	RKLPDATGR	−218.1	−225.9	−235.6
Pep4	RKLPDARGD	−217.4	−216.3	−196.6
Pep5	QPYLFATDSLIKLKA	−215.4	−224	−223.3
Pep6	GHTHYHAVRTQTQAG	−204.9	−194.6	−200
Pep7	GHTHYHAVRTQTRGD	−204.2	−185.5	−213.5
Pep8	GHTHYHAVRTQTLKA	−198.1	−181.9	−213.7
Pep9	GHTHYHAVRTQTDLN	−196.4	−213.4	−202.2
Pep10	RKLPDAVLP	−195.9	−205.3	−183.1

## Data Availability

Not applicable.

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
