# Peer review of "Peptides for Coating TiO2 Implants: An In Silico Approach"

_ijms, 2022, doi:10.3390/ijms232214048_

Round 1
Reviewer 1 Report
Aim of the present study is to find the best peptide sequence interacting with TiO2 surfaces of dental implants to improve their osseointegration. The authors selected some peptide sequences that show the best interaction,documented by most appropriate methods,as compared to the RGD as maximal interaction.However the possible transfer into the practice seems not applicable due to the high costs and complexity of sample preparation.Modification of implant surfaces by physico-chemical methods should be a simpler and cheaper way to improve implant osseointegration .
Author Response
We kindly appreciate the reviewer's suggestion. We understand the reviewer’s concern, especially regarding industrial scaling. It is important to note that the major idea of ​​this article is to reduce costs in the TiO2 surface biomimetization process. Molecular docking and dynamics simulations are currently widely used in the pharmaceutical industry, especially for drug repurposing (DOI: 10.1007/978-1-4939-8955-3_2, \https://doi.org/10.1016/j.ymeth.2022.02.004,https://doi.org/10.1016/j.bbrep.2022.101225). Despite the high cost cited by the reviewer (1mg of a synthetic peptide cost approximately 500 USD), studies on TiO2 sensitization with peptides in the literature use very low concentrations (0.0045mg/mL – 0.00045mg/mL / DOI 10.1007/s10856-011-4479-0). Therefore, a small amount of peptides can sensitize a considerable TiO2 surface. It is important to mention that we are performing further in vitro experiments. Of the peptides studied in the simulations, we chose the three best ones to carry out these assays. Thus, we saved reagents and bench time. We hope we can share these results soon enough.Reviewer 2 Report
The manuscript describes the investigation of the different bioactive peptides as bridging molecules between TiO2 implants and integrins, the cell receptors that control osteogenesis, differentiation and cell-to-cell interactions in order to select the peptides with great potential to bind the integrins.
The novelty is not apparent because there exists literature on peptide coating on titanium implants, so a novelty and significance statement is needed upfront. The method is appropriate for achieving the objectives, but lots of details and justification have been omitted. All the structures of the integrins and the proteins should be presented (majority in SI), docking sites should be presented. Some explanation as to how binding affinity, its accuracy, and the comparability of one binding score for one peptide to another peptide would be useful.
Furthermore, the results presentation needs to be improved. From Fig.1 we can derive that the peptides that were docked equilibrate at different times, but RMSD only indicates that the geometry is no longer significantly fluctuating and holds no information of relative stability. A visualization of the peptides before and after MD would be more useful for comparison of the orientation of each of the three peptides with each other. From docking simulations, one obtains other results, but these have been omitted, too.
SI is unacceptable, should be in one file, containing all the output results that are not presented in the main text.
Hence, a major revision is needed.
Other than addressing the main weakness listed above, the authors should also address the following additional points:
1. Introduction is overly brief and must be improved.
Please add the most up to date publication, “Its high biocompatibility is due to the titanium dioxide (TiO2) layer, which is formed on the titanium surface when it comes into contact with oxygen [5, 10.3390/met12030406]
Please introduce the current literature on functionalizing implants with bioactive molecules, both experimental and computational.
2. L52 ‘equate strength, toughness and corrosion and fracture resistance’ – remove the first ‘and’ in place of comma
3. L56 The explanation of implant interaction with bone (L56 onwards) should be expanded on from a biochemical perspective to explain precisely what separates a good peptide binder is in the context of integrins. Where on the integrin are the peptides binding? Is there always a specific active site?
3. L62 Some context as to how osteogenesis and differentiation are catalyzed at the cellular level may help to highlight the importance of good peptide binding.
4. L67 Will this study serve to influence/support future/current in vitro work? If so, how? Following the same logic, L241, which future in vivo studies do you recommend? How will this study guide experimental works? What have we learnt? All these remain vaguely stated.
5. L90 Both tables need to highlight the peptide with best binding affinities better so it is apparent.
6. L137 Fig.1 has too much empty space and there is no clear trend apparent from the figure alone. The resolution is rather low.
Author Response
- The manuscript describes the investigation of the different bioactive peptides as bridging molecules between TiO2 implants and integrins, the cell receptors that control osteogenesis, differentiation and cell-to-cell interactions in order to select the peptides with great potential to bind the integrins. The novelty is not apparent because there exists literature on peptide coating on titanium implants, so a novelty and significance statement is needed upfront. The method is appropriate for achieving the objectives, but lots of details and justification have been omitted. All the structures of the integrins and the proteins should be presented (majority in SI), docking sites should be presented. Some explanation as to how binding affinity, its accuracy, and the comparability of one binding score for one peptide to another peptide would be useful.
Response: We appreciate the reviewer’s comments. We added a paragraph about the novelty brought by the article (Lines 101-104) (highlighted in yellow). The PDB structures used for the simulations are mentioned in the topic 4.1 Proteins and peptides preparation. We presented all the docking results in SI, although we found unfeasible make available the PDB generated by the analysis once we performed 156 peptide-integrin docking simulations. We added in the SI1 caption of the docking analysis a paragraph about the interpretation of the scores, as suggested (highlighted in yellow).
- Furthermore, the results presentation needs to be improved. From Fig.1 we can derive that the peptides that were docked equilibrate at different times, but RMSD only indicates that the geometry is no longer significantly fluctuating and holds no information of relative stability.
Response: We added a paragraph in the Results, commenting about the criteria used to define the best complexes (Lines 127-131) (highlighted in yellow). These criteria were not based only on RMSD, but also on other parameters evaluated during the trajectory analysis of MD simulations, as we had mentioned in the Discussion (Lines 177-184). All the plots related to the MD analyses of the 10 complexes were included in SI2.
- A visualization of the peptides before and after MD would be more useful for comparison of the orientation of each of the three peptides with each other. From docking simulations, one obtains other results, but these have been omitted, too.
Response: We provided .pdb files with the overlapping best three complexes before and after MD simulation (SI3). The overlapping was made using the MatchMaker tool for structural comparison in the UCSF Chimera software. In the files, the identification “#0.1” is related to the complex in the time 0, whereas the identification “#0.2” is related to the structure in the time 100 ns. A .pdb visualization software is needed to open these files. We encourage the use of UCSF Chimera (https://www.cgl.ucsf.edu/chimera/download.html). As we mentioned, the docking sites of the 156 docking analysis would generate a very extensive SI. To simplify the results, we exposed them in Tables (SI1).
- SI is unacceptable, should be in one file, containing all the output results that are not presented in the main text.
Response: We kindly appreciate the reviewer's suggestion. We summarize the SI1 to a single file as suggested.
- Hence, a major revision is needed.
Other than addressing the main weakness listed above, the authors should also address the following additional points:
- Introduction is overly brief and must be improved.
Response: We kindly appreciate the reviewer's suggestion and made the introduction longer, mentioning the biological aspects, as requested (Lines 57-85) (highlighted in yellow).
- Please add the most up to date publication, “Its high biocompatibility is due to the titanium dioxide (TiO2) layer, which is formed on the titanium surface when it comes into contact with oxygen [5, 10.3390/met12030406]
Response: We kindly appreciate the reviewer's suggestion and added the reference (Line 46) (highlighted in yellow).
- Please introduce the current literature on functionalizing implants with bioactive molecules, both experimental and computational.
Response: We appreciate the reviewer’s comment and made the changes accordingly (Lines 224-244) (highlighted in yellow).
- L52 ‘equate strength, toughness and corrosion and fracture resistance’ – remove the first ‘and’ in place of comma
Response: We kindly appreciate the reviewer's suggestion. We removed the “and” as suggested (Line 52) (highlighted in yellow).
- L56 The explanation of implant interaction with bone (L56 onwards) should be expanded on from a biochemical perspective to explain precisely what separates a good peptide binder is in the context of integrins. Where on the integrin are the peptides binding? Is there always a specific active site?
Response: We kindly appreciate the reviewer's suggestion and aborded these topics in the Introduction (Lines 79-99)(highlighted in yellow). More on the matter is described in the Discussion (Lines 147-153).
- L62 Some context as to how osteogenesis and differentiation are catalyzed at the cellular level may help to highlight the importance of good peptide binding.
Response: We kindly appreciate the reviewer's suggestion and aborded these topics in the Introduction (Lines 60-79)(highlighted in yellow).
- L67 Will this study serve to influence/support future/current in vitro work? If so, how? Following the same logic, L241, which future in vivo studies do you recommend? How will this study guide experimental works? What have we learnt? All these remain vaguely stated.
Response: We appreciate the reviewer’s comment and made the changes accordingly (Lines 244 and 245) (highlighted in yellow).
- L90 Both tables need to highlight the peptide with best binding affinities better, so it is apparent.
Response: We highlighted the peptide with the best binding affinity in the Tables, accordingly (Tables 1 and 2)(highlighted in yellow).
- L137 Fig.1 has too much empty space and there is no clear trend apparent from the figure alone. The resolution is rather low.
Response: We kindly appreciate the reviewer's suggestion. We adjusted the image so it can be more compact and enhanced its quality (Figure1).
Round 2
Reviewer 2 Report
The quality improvement is negligible after the revision, in that the authors refused to present any visualization of the docking complexes in the manuscript. With a few data tables and carelessly-prepared plots that do not mean much without careful reading the text for uninitiated readers, the paper is boring, uninspiring, and chance of attracting much interest and citations is low. The motivation of insisting on such boring representation is unknown, unless authors plan on presenting the more exciting results in other publications.
Here, a quick literature search returned this quality paper, please read and learn what a typical modelling and docking analysis work should present.
J. Med. Chem. 2006, 49, 984-994.
At least present the 3 integrins, and some selected integrin-protein docking and MD images before and after simulation, in the main text.
The highlighted text does help novelty statement, leave as is.
The multiple SI files are not appreciated. Why not put all the supporting information in one file to support the study? This will greatly reduce reviewers’ and readers’ time in finding useful info. What is peptide 1-10? What is each sequence in SI, is not clear.
Fig. 1, the blank space within each plot should be greatly reduced by adjusting the axis ranges, e. g. a, first panel, change RMDS range to 0-1, and so on.
Author Response
- The quality improvement is negligible after the revision, in that the authors refused to present any visualization of the docking complexes in the manuscript. With a few data tables and carelessly-prepared plots that do not mean much without careful reading the text for uninitiated readers, the paper is boring, uninspiring, and chance of attracting much interest and citations is low. The motivation of insisting on such boring representation is unknown, unless authors plan on presenting the more exciting results in other publications.
Here, a quick literature search returned this quality paper, please read and learn what a typical modelling and docking analysis work should present.
- Med. Chem. 2006, 49, 984-994.
Response: We understand the reviewer’s concern. Although, we kindly disagree that the tables and figures available are boring. Since the focus of the article is to serve as a basis for future in vitro tests, we strongly believe that tables summarize well our results (the peptides that we found to be promising). Sometimes, simple is more informative.
About the reviewer’s first suggestion: “All the structures of the integrins and the proteins should be presented (majority in SI), docking sites should be presented.” Our reluctance in bringing the results of the analyzes in figures was also due to the large number of figures generated per peptide. Specially, in the conformation requested by the reviewer (integrins alone and conjugated with peptides before and after the MD analyzes). Figure 2 (regarding a single peptide), for example, contains 12 images. We modeled 52 peptides and performed 156 docking analyzes in this study. This would generate 624 images. We also fear the online submission system would not be able to support the size of the file containing all these images in a proper definition. That is why we do not think it should be in the manuscript or in the SI (and preferred to use tables to disclosure our results), not because we wanted keep data to publish it elsewhere.
Besides, in our opinion, figures are most welcome and should be explored in manuscripts that discuss more specific interactions (between residues, for example, as in the above reference suggested by the reviewer).
Apart from that, another aim of this study is to spread and encourage the bioinformatics approach used (a flowchart of the work had always been available) so that other researchers could use it in their respective works, saving bench time. We reaffirm that we have no intention of publishing the results displayed here elsewhere, after all we made the .pdb files of the docking results available (as SI) when requested in the first round of reviews.
- At least present the 3 integrins, and some selected integrin-protein docking and MD images before and after simulation, in the main text.
Response: We added a figure in the manuscript (Figure 2), as requested.
- The highlighted text does help novelty statement, leave as is.
Response: We appreciate the reviewer’s kind comment. We left the text as it is. We only had to add a caption for Figure 2, as well as a phrase regarding the figure in the text (LINE) (highlighted in green).
- The multiple SI files are not appreciated. Why not put all the supporting information in one file to support the study? This will greatly reduce reviewers’ and readers’ time in finding useful info. What is peptide 1-10? What is each sequence in SI, is not clear.
Response: We appreciate the reviewer’s comment. Again, our concern in making one single file is due to the size of the single .pdf file which would contain 4 tables and 38 graphics. Despite that, we summarized the SI1 and SI2 into one SI (now SI1), in accordance with the reviewer's wishes. We also added a caption to explain the graphics better. Although, we were not able to summarize SI3 (now SI2), once they are .pdb files.
- Fig. 1, the blank space within each plot should be greatly reduced by adjusting the axis ranges, e. g. a, first panel, change RMDS range to 0-1, and so on.
Response: We appreciate the reviewer’s comment. We did it accordingly and adjusted the ranges of the graphics in Figure1, as well as in SI.
Sincerely,
Almerinda Agrelli
Round 3
Reviewer 2 Report
The authors have addressed all concerns and improved the overall presentation and quality.